# ADVANTAGE-CONDITIONED DIFFUSION: OFFLINE RL VIA GENERALIZATION

## ABSTRACT

Reinforcement learning algorithms typically involve an explicit maximization step somewhere in the process. For example, policy gradient methods maximize an estimate of the expected return, and TD methods maximize the target value while training a critic network. However, explicit maximization of neural function approximators leads to learning out-of-distribution actions during offline training, which in turn can lead to overestimation and distributional shift of the learned policy. Can we instead devise an offline RL method that maximizes the value implicitly, via generalization? In this paper, we show how expressive conditional generative models combined with Implicit Q-Learning backups can enable this, providing an offline RL method that attains good results through generalization alone, and state-of-the-art results when combined with a simple filtering step that maximizes over samples from the policy only at evaluation time. We believe that our work provides evidence that the next big advancements in offline RL will involve powerful generative models.

## 1  INTRODUCTION

Offline reinforcement learning (offline RL) holds the promise of leveraging large datasets for learning-based control and decision making without costly online exploration (Levine et al., 2020). However, standard algorithms typically involve iteratively choosing actions that maximizes the estimated value of a neural-network critic (*e.g.*, as with Q-learning or Q-function actor-critic methods). As this critic is imperfect, the maximizing action will often have an erroneous (and overestimated) value prediction (Fujimoto et al., 2019; Kumar et al., 2020; Levine et al., 2020). While in the online setting an agent that learned such an action may collect more data and correct itself, in offline RL these mistakes cannot be reversed. Therefore, a variety of modifications to standard methods have been proposed specifically to avoid out-of-distribution actions (Fujimoto et al., 2019; Kumar et al., 2019a; 2020). One may argue, however, that learning methods that are based on explicit maximization are simply not fit for the offline setting, further implying that supervised learning (SL)-based methods should be explored. Indeed, instead of maximizing estimated returns, an agent may learn the relation between outcomes of trajectories (datum $X$ in SL) and actions that led to them (label $Y$), so that a policy is conditioned on those outcomes at test time. Following this motivation, RvS (RL via SL)—or Upside-Down RL (Schmidhuber, 2019)—methods suggest to view offline RL as maximum likelihood estimation conditioned on a desired return, or even advantage value (Kumar et al., 2019b; Emmons et al., 2021). Surprisingly, such simple algorithms, without employing any critic networks, can improve on standard behavioral cloning. Nevertheless, their performance does not match the state of the art, which should not be surprising. Clearly, it is difficult to specify the optimal in-distribution return value, to condition the policy on, that it could obtain at test time while temporaly decomposing it across encountared states in a trajectory. Lastly, Brandfonbrener et al. (2022) have shown formally that to produce reliable return-conditioned policies, RvS requires data coverage of these returns. Can we circumvent these problems and devise RvS policies that are not sensitive to the conditioning values?

Surprisingly, we can construct an optimal policy by conditioning the behavior-cloning distribution on zero. To understand this puzzle, recall that the advantage value of optimal actions under the optimal Q-function is always zero (Sutton & Barto, 2018), regardless of their actual return. While having only finite offline data we cannot compute optimal Q-functions, one can learn approximately optimal ones for in-distribution actions with Implicit Q-Learning (Kostrikov et al., 2021, IQL). An action

distribution that is conditioned on such state-advantage pairs could then be used to extract the optimal policy at test time by conditioning it on zero advantage. In effect, conditioning on advantages puts temporal compositionality back into the conditional policy training, enabling it to leverage generalization to maximize the value without actually needing to extrapolate to unseen returns. Based on this reasoning, we propose a method that *(1)* precomputes the advantage-value labels for state-action pairs in the dataset with IQL, and then *(2)* trains a behavioral-cloning policy conditioned on state and these advantage values, such that *(3)* at test time, leveraging the approximate optimality of our value functions, we can extract a strong policy from this model by conditioning it on zero advantage at every state. We implement a few variants of this approach, the best one employing diffusion models, as they have proven successful at capturing fine conditioning information and representing highly complex distributions (Ho et al., 2020; Song et al., 2020; Ho & Salimans, 2022). We evaluate our method—*Advantage-Conditioned Diffusion* (ACD)—on the D4RL benchmark tasks (Fu et al., 2020), and show that by itself this method can deliver satisfying results without any explicit maximization, and in combination with additional filtering techniques it can reach state-of-the-art performance on the challenging antmaze tasks.

## 2  RELATED WORK

Many of the recently proposed offline RL methods incorporate some sort of penalty, regularizer, or constraint that avoids overestimation of out-of-distribution actions, so that standard algorithmic techniques that, unlike our ACD, involve explicit maximization could be applied and learn only in-distribution behaviors (Wu et al., 2019; Kumar et al., 2020; Fujimoto & Gu, 2021). Departing from this perspective, Implicit Q-Learning (Kostrikov et al., 2021, IQL) trains a value function that accomplishes this objective implcitly via the expectile loss, and provides value estimates based on in-distribution actions. Nevertheless, extracting a good policy from IQL critics remains an opem question. The original method trains a unimodal state-conditioned policy with Advantage-Weighted Regression (Peng et al., 2019), which has that flaw that it incentifies the policy to learn all actions in the dataset, including the most suboptimal ones. In contrast, we use the advantages to form labels to condition our diffusion policy on, which we train with the denoising loss. With such a model, we can mitigate the influence of poor actions at test time by conditioning the policy on high advantage values.

Another line of research, sometimes referred to as Upside-Down RL (Srivastava et al., 2019, UDRL) or RvS (Kumar et al., 2019b; Emmons et al., 2021), observes that any dataset that consists of suboptimal trajectories may provide an optimal supervision for conditional maximum-likelihood models. For example, the original UDRL papers (Schmidhuber, 2019; Srivastava et al., 2019) suggest that the policy can be conditioned on both the reward and the time required to earn it, and demonstrate preliminary results in OpenAI Gym (Brockman et al., 2016) environnments. Reward-conditioned policies (RCP) of Kumar et al. (2019b), in the offline setting, learns the value function of the behavior policy, and then conditions the test-time policy on the desired advantage value. In order to achieve performance better than the bahavior policy, this value must be greater than zero and tuned. This approach induces a trade-off: zero advantages induce in-distribution actions with behavior policy-like performance, while higher advantages can improve that performance, but are likely to fall in the trap being queried for out-of-distribution values. Indeed, the resulting method was found to perform worse than AWR (Peng et al., 2019), which we benchmark our method against. Our algorithm avoids the above problems by using IQL (Kostrikov et al., 2021) advantage functions, which have that property that actions with zero advantage value are nearly-optimal and in-distribution. Therefore, at test time, we can safely condition on zero advantage and achieve good performance. Another work that uses rewards for direct supervision is that of *Decision Transformer* (Chen et al., 2021, DT), where the policy is extracted from a transformer (Vaswani et al., 2017) that models the trajectory as a sequence. However, for this model to perform well it is necessary to condition it on a long (up to 50 steps) subset of the action-observation-reward history which incurs additional computational burden on top of that of storing and training the model. Even at this expense, the model was found not to perform well in more challenging tasks, like antmaze in D4RL (see, for example, evaluations in Kostrikov et al. (2021)). In contrast, our method, apart from the scalar advantage, conditions the policy only on the current state and achieves good results.

Recently, we have observed a surge in interest of applying *Diffusion Models* (Sohl-Dickstein et al., 2015; Ho et al., 2020; Song et al., 2020) in RL. Diffusion Q-Learning (Wang et al., 2022, DQL)

models the policy with a diffusion model and optimizes it against both the maximum-likelihood and policy-gradient losses. A possible drawback of this algorithm is that it requires backpropagation through the entire denoising process which introduces additional computational overhead. Implicit Diffusion Q-Learning (Hansen-Estruch et al., 2023, IDQL), on the other hand, optimizes the diffusion model solely with respect to the denoising loss, without the need for backpropagation through the denoising process. Unfortunately, this training scheme does not put any preference on performant actions, and in order to obtain good performance at test time, one must sample multiple actions and execute the one with the highest IQL value. Our ACD, instead, incorporates information from IQL critics into the diffusion training, so that at test time we can bias the model's distribution towards high-quality actions.

## 3 PRELIMINARIES

This section explains the necessary background for this paper. We recall the foundations of offline RL, review the main idea behind RL via supervised learning, and summarize diffusion models.

### 3.1 OFFLINE REINFORCEMENT LEARNING

We consider a *Markov decision process* (Sutton & Barto, 2018, MDP) where, at timestep $t \in \mathbb{N}$, an agent is at state $s_t \in \mathcal{S}$, takes an action $a_t \in \mathcal{A}$, and moves to the next state $s_{t+1} \sim P(\cdot|s_t, a_t)$, collecting along a reward $r(s_t, a_t)$. The agent's goal is to find a state-conditioned policy $\pi$ that maximizes the expected *return*,

$$\eta(\pi) = \mathbb{E}\Big[ \sum_{t=0}^{\infty} \gamma^t r(s_t, a_t) \mid \pi \Big],$$

sometimes referred to as *expected total reward*, for a predefined value $\gamma \in [0, 1)$. The maximizer policy of the above equation induces the action-value function that equals the expected return given the state and action taken, and satisfies the Bellman optimality equation

$$Q^*(s, a) = r(s, a) + \gamma \cdot \mathbb{E}_{s' \sim P}\Big[ \max_{a' \in \mathcal{A}} Q^*(s', a') \Big]. \tag{1}$$

In the infinite data regime, this function can be found with approximate fixed point iteration methods which train the left side of the above equation to match the right one (Sutton & Barto, 2018; Mnih et al., 2015; Fujimoto et al., 2018). The policy can be then trained to maximize the value function. However, in offline RL, the agent cannot interact with the MDP directly, but instead is given a dataset of $N$ transitions $\{s_{t_i}, a_{t_i}, r(s_{t_i}, a_{t_i}), s_{t_i+1}\}_{i=1}^N$. This setting prevents application of standard RL algorithms which update the policy with stochastic gradient ascent since actions that optimize the estimated value function may not be covered by the dataset's distribution, and thus their optimality can be an error of the estimate of the return (Levine et al., 2020). Implicit Q-Learning (Kostrikov et al., 2021, IQL) attempts to solve this problem by replacing the max operator at the next state in Equation (1) with an *implicit*, in-distribution max. To that end, it trains state-value and action-value function pair $(V^\tau, Q^\tau)$ so that $V^\tau$ models the $\tau^{th}$ expectile of $Q^\tau$ at a given state

$$V^\tau(s) = E_{a \sim \mathcal{D}|s}^\tau[Q(s, a)], \forall s \in \mathcal{S},$$

where, $\tau \in [0.5, 1)$ and $\mathbb{E}_{x \sim p}^\tau[x]$ is the expectile operator for $x \sim p(x)$. That is, the solution to

$$\mathbb{E}_{x \sim p}^\tau[x] = \arg\min_{m^\tau} \mathbb{E}_{x \sim p}\big[ \big( \tau \cdot \mathbf{1}_{(x-m^\tau \geq 0)} + (1 - \tau) \cdot \mathbf{1}_{(x-m^\tau < 0)} \big) \cdot (x - m^\tau)^2 \big],$$

where $\mathbf{1}_{(E)}$ is the indicator function of the event $E$. As $\tau \geq 1 - \tau$, the minimizer $m^\tau$ tries to make the event $x - m^\tau \geq 0$ happen rarely, and thus concentrates around higher values, converging to the maximum of the dsiribution as $\tau \to 1$. Such $V^\tau$ is then used in the Bellman backup for $Q^\tau$ so that they satisfy

$$Q^\tau(s, a) = r(s, a) + \gamma \cdot \mathbb{E}_{s' \sim P}\big[ V^\tau(s') \big].$$

As the expectile is applied over actions in the dataset's distribution only, the critic can avoid the extrapolation error (Fujimoto et al., 2019) and learn an approximately optimal value function. To

extract the policy from these critics, IQL uses Advantage-Weighted Regression (Peng et al., 2019, AWR), which is a weighted form of maximum likelihood estimation:

$$\pi^* = \arg\max_\pi \mathbb{E}_{(s,a)\sim\mathcal{D}}\big[\exp\big(\beta \cdot A^\tau(s,a)\big) \cdot \log \pi(a|s)\big],$$

where $\beta \in \mathbb{R}^+$ and $A^\tau(s,a) = Q^\tau(s,a) - V^\tau(s)$ is the IQL advantage value. Unfortunately, as AWR assigns a positive weight to every action, neural network policies are incentified to learn all of them, instead of choosing only the best behaviors. Our method tackles this problem by using the advantage as a learning signal, rather than a weight, allowing us to query the model for optmial actions at test time by specifying the advantage value.

## 3.2 REINFORCEMENT LEARNNIG VIA SUPERVISED LEARNING

The premise of Reinforcement Learnnig Via Supervised Learning (RvS) is that, by conditioning the policy on outcomes of trajectories during training, we can query it for actions that lead to a desired outcome at test time (Emmons et al., 2021). Frequently chosen outcomes include goal states (Ding et al., 2019) and the return (Kumar et al., 2019b; Chen et al., 2021). In this work, we focus on the latter as the goal-conditioned RL can be viewed as RL with an indicator reward function. To train a return-conditioned policy, one can maximize the log-likelihood

$$\mathcal{L}_{\text{RvS}}(\pi) = \mathbb{E}_{(s,a,R)\sim\mathcal{D}}\big[\log \pi(a|s, R)\big],$$

where $R$ is the reward-to-go for the state-action pair $(s, a)$. To compute the reward-to-go $R$ for a pair $(s, a)$, one can simply sum (and discount) the current and all future rewards within a trajectory in the dataset until the next $done$ signal is encountared. Then, at test time, one can obtain a performant policy by conditioning on a desired (high) return, $\pi(a|s, R_*)$, and updating $R_*$ every time by subtracting the recent reward, $R_* \leftarrow R_* - r(s,a)$. Unfortunately, although elegant, this type of methods is still behind the state-of-the-art algorithms (Emmons et al., 2021), which should not come across as surprising: naturally, to obtain returns that are better than those in the dataset, one must condition the policy on higher returns. The policy network has not seen such returns during training, and thus is likely to output poor actions. In fact, Brandfonbrener et al. (2022) formalize this intuition by deriving statistical bounds on the performance of RvS with respect to the return coverage in the dataset. According to their analysis, without data coverage of the desired returns, a good performance of the trained policy cannot be guaranteed. In contrast, this paper introduces an algorithm employing value-based methods to leverage temporal compositionality of returns, and results in a policy that outputs high-quality actions by conditioning on the—always in-distribution—zero advantage. In this paper, we will introduce a method that avoids querying for out-of-distribution actions by conditioning on the advantage function, which has the property that in-distribution returns that it models are concentrated around zero.

## 3.3 DIFFUSION MODELS

Denoising diffusion probabilistic models (Sohl-Dickstein et al., 2015; Ho et al., 2020; Song et al., 2020, DDPM) fit a data distribution $p_\theta(\mathbf{x}) \approx p_{\text{data}}(\mathbf{x})$ by reversing a *diffusion process*. That is, a process in which a point $\mathbf{x}^0 \sim p(\mathbf{x}^0)$ gets gradually blurred with Gaussian noise:

$$\mathbf{x}^t = \sqrt{1 - \beta^t}\mathbf{x}^{t-1} + \sqrt{\beta^t}\epsilon^t, \ \epsilon^t \sim N(0, I), \ t = 1, \dots, T$$

where $\beta^t$ is an increasing sequence of positive scalars, known as *variance schedule*. Note that, in this paper, we use superscripts to denote diffusion steps and substricts to denote steps in the MDP. Having learned to predict the noise vectors with denoising score matching (Vincent, 2011), DDPMs can generate new data by drawing a noise vector $\mathbf{x}^T \sim N(0, I)$ and simulating a *reverse process*:

$$\mathbf{x}_{t-1} = \frac{1}{\alpha^t}\Big(\mathbf{x}_t - \frac{\beta^t}{\sqrt{1 - \hat{\alpha}^t}}\epsilon_\theta(\mathbf{x}_t, t)\Big) + \sqrt{\beta^t}\hat{\epsilon}^t, \ \hat{\epsilon}^t \sim N(0, I), \ t = T, \dots, 1. \tag{2}$$

where $\alpha^t = 1 - \beta^t$ and $\hat{\alpha}_t = \prod_{k=1}^t \alpha_k$. One can consider an additional variable, $\mathbf{y} \sim p_{\text{data}}(\mathbf{y})$ and modeling the conditional distribution $p_\theta(\mathbf{x}|\mathbf{y}) \approx p_{\text{data}}(\mathbf{x}|\mathbf{y})$ by conditioning the noise network on $\mathbf{y}$, that is $\epsilon_\theta(\mathbf{x}^t, t, \mathbf{y})$, which was trained on joint samples $(\mathbf{x}, \mathbf{y}) \sim p_{\text{data}}(\mathbf{x}, \mathbf{y})$.

Conditional generation, most of the time, deals with $\mathbf{y}$ being categorical variables, known as classes (Russakovsky et al., 2015; Van den Oord et al., 2016; Odena et al., 2017). Nevertheless, the conditional distribution $p_{\text{data}}(\mathbf{x}|\mathbf{y})$ is well-defined even if $\mathbf{y}$ is a continuous scalar. In the next section, we leverage this property by implementing our advantage-conditioned policy as a diffusion model.

## 4 METHOD

In this section, we carefully explain how we develop our method, analyzing its key design choices one at a time.

### 4.1 OPTIMIZATION THROUGH CONDITIONING

Our derivation begins with an observation that, under the optimal action-value and state-value functions, $Q^*(s, a)$ and $V^*(s)$, the optimal action for every state has advantage value

$$A^*(s, a) = Q^*(s, a) - V^*(s) = Q^*(s, a) - \max_{a' \in \mathcal{A}} Q^*(s, a') = 0.$$

Hence, in the infinite data regime, we could learn these value functions and use them to label the state-action pairs with their advantage value. Then, we could use these labels to fit a conditional distribution $\pi(a|s, A)$. At test time, in order to extract the optimal policy from $\pi$, one can simply condition it on $A = 0$, for every state, as we formalize in the following theorem.

**Theorem 1.** *Let $\mathcal{P}$ be the set of distributions over $\mathcal{A}$ that are conditional on pairs $(s, x) \in \mathcal{S} \times \mathbb{R}$, and $d(s, a)$ be a strictly positive distribution over state-action pairs. Let $A^*(s, a) = Q^*(s, a) - V^*(s)$ be the optimal advantage function and*

$$\pi^* \in \arg\max_{\pi \in \mathcal{P}} \mathbb{E}_{(s,a) \sim d} \big[ \log \pi \big( a|s, A^*(s, a) \big) \big].$$

*Then, $\pi^*(a|s) \triangleq \pi^*(a|s, 0)$ is the optimal policy.*

*Proof.* First, note that an action $a$ is optimal at state $s$ if and only if $A^*(s, a) = 0$, since this is equivalent to $Q^*(s, a) = V^*(s) = \max_{a' \in \mathcal{A}} Q^*(s, a')$. Further, $\pi^*(a|s, A^* = 0) > 0$ implies that $\Pr\big(A^*(s, a) = 0|s, a\big) > 0$, and since $A^*(s, a)$ is a deterministic mapping, this means that $A^*(s, a) = 0$. Hence, $a$ is optimal at state $s$, which finishes the proof. $\square$

Of course, in offline RL, we only have access to a finite dataset, implying that the optimal value functions cannot be computed. Nevertheless, we can learn *approximate, in-distribution* value functions $V^\tau$ and $Q^\tau$ with IQL (Kostrikov et al., 2021). As $\tau \to 1$, $V^\tau(s)$ converges to the max of $Q^\tau(s, a)$ over in-distribution actions, and actions with $A^\tau(s, a) = 0$ converge to the optimal action at state $s$. Thus, we can approximately recover the policy from Theorem 1 by computing the advantage labels $A$ for $(s, a) \in \mathcal{D}$ with IQL, and use them to learn the conditional distribution $\pi^*(a|s, A)$. This gives rise to a dataset

$$\mathcal{D}_{\text{labeled}} = \big\{ s_{t_i}, a_{t_i}, y_{t_i} = A^\tau(s_{t_i}, a_{t_i}) \big\}_{i=1}^N \tag{3}$$

Having obtained such a dataset, we train a neural-network action distribution conditioned on the state and advantage value. Then, at test time, we deploy the policy by conditioning the distribution on $A = 0$. Of course, it may be tempting to condition on higher advantage values. However, actions with substantially greater values are not guaranteed to be in-distribution. We illustrate this by plotting the histogram of IQL advantage values in antmaze-medium-play in Figure 1, which reveals that the coverage of advantages suddenly drops above 0. Therefore, while it is possible to increase that value from zero to marginally improve the performance and not go out of distribution, the value that allows for that will depend on the task and the dataset. Hence, we simply condition on $A = 0$ at test time.

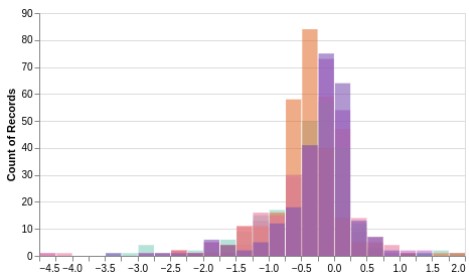

Figure 1: The union of histograms, with bin width of 0.25, of IQL advantage values in antmaze-medium-play. Actions with positive advantage have a very low coverage.

The policy can be implemented in a number of ways, but the most typical one for RL is a Gaussian distribution parameterized by multi-layer percepteron (Schulman et al., 2015; Haarnoja et al., 2018; Kumar et al., 2020). We implement our approach with this architecture for a proof of concept

| Dataset | SAC | BC | AWR | ACG |
|---|---|---|---|---|
| antmaze-umaze | 0 | 65 | 56 | **81.9** |
| antmaze-umaze-diverse | 0 | 55 | **70.3** | 57.9 |
| antmaze-medium-play | 0 | 0 | 0 | **14.1** |

Table 1: ACG outperforms Soft Actor-Critic (SAC), Behavior Cloning (BC), and Advantage-Weighted Regression (AWR) in 2 out of 3 tested D4RL tasks, and places second once.

of optimization via generalization with IQL advantage values. We test this algorithm (which we refer to, for now, as ACG: Advantage-Conditioned Gaussian) against three approaches: off-policy RL (represented by SAC (Haarnoja et al., 2018)), maximum-likelihood estimation (represented by BC), and weighted maximum-likelihood (represented by AWR (Peng et al., 2019)), in three D4RL environments, as presented in Table 1. As oppose to those methods, ACG was capable of improving upon the behavior policy in all three tasks, placing first in 2 out of 3 of them, and once coming second.

Nevertheless, Gaussian distributions parameterized by MLPs are not the most effective probabilistic models. Thus, we continue developing our method by incorporating recent advances from generative modeling—diffusion models (Ho et al., 2020).

## 4.2 ADVANTAGE-CONDITIONED DIFFUSION

In this subsection, we combine the idea of policies conditioned on optimal advantages with the diffusion model toolkit. We implement the conditional distribution $\pi_\theta(a|s, A)$ as a diffusion model, which we implement with a noise network $\epsilon_\theta(\mathrm{a}^t, t, \mathrm{s}, A)$. That is, the network that generates an action $\mathrm{a} = \mathrm{a}^0$ from random noise $\mathrm{a}^T \sim N(0, I)$ and takes the state and advantage information as input. Furthermore, inspired by *classifier-free guidance* (Ho & Salimans, 2022), we additionally train an (advantage-)unconditional noise model $\epsilon_\theta(\mathrm{a}^t, t, \mathrm{s})$. In practice, we train only one network, and call the unconditional model by passing a special null token $\emptyset$ into the network, $\epsilon_\theta(\mathrm{a}^t, t, \mathrm{s}, \emptyset)$. The token is implemented by prepending an indicator variable to the advantage, resulting in an input vector $\mathrm{y} = [\mathbf{1}_{(\text{not\_null})}, \mathbf{1}_{(\text{not\_null})} \cdot A] \in \mathbb{R}^2$. As such, during training, we drop the advantage information such that we have $\mathbf{1}_{(\text{not\_null})} = 0$ with probability $p = 0.1$. After training the model, at test time, we sample from it using a linear combination of the score networks,

$$\epsilon_\theta^\omega(\mathrm{a}^t, t, \mathrm{s}, A = 0) \triangleq (1 + \omega) \cdot \epsilon_\theta(\mathrm{a}^t, t, \mathrm{s}, A = 0) - \omega \cdot \epsilon_\theta(\mathrm{a}^t, t, \mathrm{s}, \emptyset), \tag{4}$$

where $\omega \in \mathbb{R}$. It is worth noting that for $\omega = -1$, the model becomes fully unconditional, and for $\omega = 0$ the linear-combination trick is identical to a vannila conditional network. In our experiments, we found that this technique can significantly boost performance (*e.g.*, from 43.2 to 56.1 on antmaze-medium-play), and throughout the paper we work with $\omega = 3$. We summarize the whole method in Algorithm 1.

---

**Algorithm 1** Advantage-Conditioned Diffusion

1: **Input:** dataset $\mathcal{D}$, critic networks $Q_\phi(s, a)$ & $V_\phi(s)$, noise network $\epsilon_\theta$, Bernoulli parameter $p$.
2: **for** step $= 0, 1, \dots$ **do**
3:     Sample a batch $\mathcal{B} \subset \mathcal{D}$.
4:     Make an IQL update on $Q_\phi$ & $V_\phi$.
5:     $\forall (s, a) \in \mathcal{B}$, label them with $A_\phi(s, a) = Q_\phi(s, a) - V_\phi(s)$.
6:     Mask the advantage labels independently at random with probability $p$.
7:     Make a gradient descent step on $\epsilon_\theta$ with the denoising loss.
8: **end for**
9: **Deploy:** Diffusion policy parameterized by $\epsilon_\theta^\omega(\mathrm{a}^t, t, \mathrm{s}, A = 0)$ from Equation (4).

---

## 4.3 ADVANTAGE REPRESENTATION

We elaborate that we explored other forms of conditioning of the advantage value which we discuss here. First, while working on ACG, we tried discretizing the real axis into $N_{\text{bin}}$ bins, and

encode an advantage value with the $i^{th}$ standard basis vector if it belonged to the $i^{th}$ bin. This indeed improved the network's performance on some tasks. We report example results in Table 2.

Nevertheless, we do not include more results for this technique since we found that our final method—ACD—works better with raw advantage values. Furthermore, poor results for ACG with scalar advantages can be improved by changing the optimizer from Adam (Kingma & Ba, 2014) to AdamW (Loshchilov & Hutter, 2017). Such an observation provides an interesting insight into the difference between the abilities of MLPs and DDPMs: the less expressive MPLs struggle to generalize to information from a single continuous signal but work descently with crude, discrete signals, while the more expressive diffusion model benefits from exact continuous advantage values.

| Task | Raw | Discrete |
|------|-----|----------|
| antmaze-umaze | 81.9 | **82.1** |
| antmaze-medium-play | 14.1 | **34.9** |

Table 2: Ablation of advantage representation (raw real values vs discretized categorical) for ACG in antmaze tasks.

While working with ACD, we tried the previously-mentioned raw and discrete representations, as well as more sophisticated methods of modeling the advantage input.

Namely, we attempted to use Transformer's sinusoidal encoding (Vaswani et al., 2017) of advantage, as well as to learn an MLP mapping from raw values to a vector that would be an input to the DDPM. While we found these two approaches perform similarly, they significantly worsened the performance of our policy, so we decided to work with raw scalars.

| Raw | Discrete | Sinusoidal | Learned |
|-----|----------|------------|---------|
| **53.4** | 36.5 | 26.8 | 27.7 |

Table 3: Ablation of advantage representations for ACD in antmaze-medium-play tasks.

We also tried training an unconditional DDPM and sample from it in a *classifier guidance* style—by incorporating the gradient of the Q-function with respect to the denoised action, $\nabla_{a^t} Q^\tau(s, a^t)$, into the reverse process, similarly to Diffuser (Janner et al., 2022). However, in addition to being more computationally expensive due to the gradient of the Q-function, this variation performed worse than classifier-free guidance in our early experiments. Hence, we decided not to pursue it further, and leave a more exhaustive exploration of it as future work.

## 4.4 VALUE FILTERING

While we will show that the method described above is already sufficient to attain good performance for offline RL without any explicit maximization, we found that in order to obtain results that are competitive with state-of-the-art methods it is beneficial to add an additional test-time *value filtering* phase as a post-processing step on our method, similarly to the technique used by IDQL (Hansen-Estruch et al., 2023). At test time, we sample $N_{\text{sample}}$ actions from the conditional DDPM,

$$\left\{ a^{(i)} \sim \pi_\theta\left(a^{(i)}|s, A=0\right) \right\}_{i=1}^{N_{\text{sample}}},$$

compute their Q-values, and execute the action with the largest Q-value. We refer to this method as *filtered* ACD (ACD-F). While this does introduce explicit maximization, this is only performed at test time, and not during training. Note that we have not tuned the value of $N_{\text{sample}}$, but simply adopted it from IDQL. This may have impact of the performance of ACD-F, since the goal of conditioning is to narrow down the distribution of sampled actions (center it around a particular mode). Thus, it is likely that the same number of samples that was optimal for IDQL may be too high for ACD, increasing the probability of encountering an adversarial action. Indeed, we noticed a drop in performance of ACD-F with respect to ACD in antmaze-umaze-diverse (see Table 4). Nevertheless, we keep this value for simplicity, and leave its analysis and tuning for future work.

## 5 EXPERIMENTS

To implement the diffusion policy, we borrow the core of our architecture from IDQL (Hansen-Estruch et al., 2023) which consists of three residual blocks with layer normalization. Advantage (or null) conditioning information is done by concatenating the 2-dimension vector y from Subsection 4.2 to the state vector. During training, we choose to drop the advantage information and train

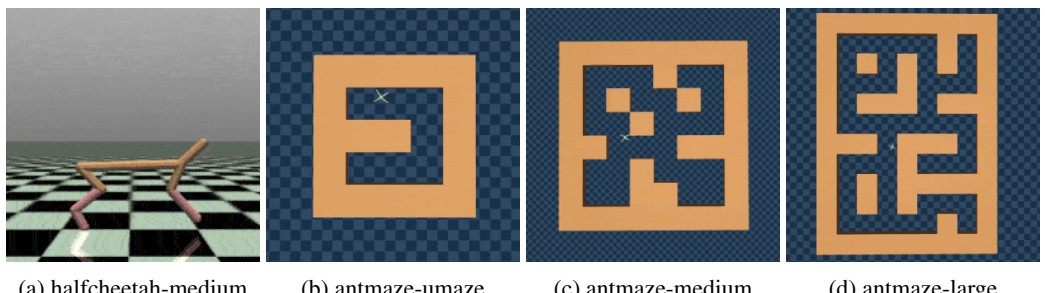

| (a) halfcheetah-medium | (b) antmaze-umaze | (c) antmaze-medium | (d) antmaze-large |

Figure 2: Visualization of D4RL tasks that we evaluate on.

| Dataset | BC | AWR | IQL | DQL | IDQL | ACD | ACD-F |
|---|---|---|---|---|---|---|---|
| halfcheetah-med | 44 | 47.4 | **51.1** | **51.0** | 48.5 | 49.2 | **51.6** |
| antmaze-umaze | 65 | 56 | 86.4 | 47.6 | **93.8** | 78.4 | 77.1 |
| antmaze-umaze-div | 55 | **70.3** | 62.4 | 35.8 | 62 | 59 | 45.8 |
| antmaze-med-play | 0 | 0 | 76 | 42.5 | **86.6** | 56.1 | 83.5 |
| antmaze-med-div | 0 | 0 | 70.0 | 78.6 | 84.8 | 56.3 | **86.8** |
| antmaze-large-play | 0 | 0 | 39.6 | 46.4 | **62** | 14.7 | 57.6 |
| antmaze-large-div | 0 | 0 | 47.5 | 57.3 | **51.8** | 11.7 | **51.1** |

Table 4: Performance comparison of a range of offline RL methods, including state-of-the-art IQL- and DDPM-based methods. ACD achieves good results with advantage-conditional generation only, without explicit maximization at all. With value filtering, ACD-F is competitive with SOTA.

the null model with probability $p = 0.1$. At test time, we sample from the model with classifier-free guidance (Eq. (4)) with $\omega = 3$. We evaluate our methods, ACD and ACD-F, on challenging D4RL tasks and compare to prior methods: Behavior Cloning (BC), Advantage-Weighted Regression (Peng et al., 2019, AWR), Implicit Q-Learning (Kostrikov et al., 2021, IQL), Diffusion Q-Learning (Wang et al., 2022, DQL), and Implicit DQL (Hansen-Estruch et al., 2023, IDQL).

We focus, in particular, on antmaze tasks (Figures (2b)-(2d)), for which the datasets were created running different goal-conditioned policies. These tasks test the agent's *stitching* ability—composing optimal behavior in different regions of the state space—that allows it to generate better trajectories than those of the behavior policy. As we can see in Table 4, ACD by itself is sufficient to improve upon the behavior policy and significantly outperforms AWR. This confirms that it is possible to learn competitive policies for offline RL without explicit maximization and instead conditional generation only. Unsurprisingly, additional samples and value filtering improves the performance of our method, making ACD-F significantly better than prior art, apart from IDQL, with which it is competitive in nearly all tested tasks.

## 6    CONCLUSION

In this paper, we presented a method of solving offline reinforcement learning with conditional generative modeling. The conditioning variables of our policy are optimal advantage values, which have the property of equaling zero at optimum. In order to learn approximately optimal advantages for in-distribution actions, we use Implicit Q-Learning (IQL), and to extract performant behavior from the learned model, we simply condition it on the zero scalar. This procedure does not require explicit maximization steps in the policy training. We investigated the performance of many design choices, including network architectures, training, training, and sampling steps. Our final policy itself, which is trained to learn conditional distributions, is implemented with a diffusion model, from which we sample using a classifier-free guidance-style method. The results that we obtained prove that offline RL policies can be trained with generative losses, without the need of explicit maximization of the return, and deployed with a simple conditioning scheme. Out future work will focus on a deeper exploration of the design choices in diffusion models that allow for sampling the highest-quality actions without the need of excessive sampling and value filtering. We anticipate and look forward to seeing generative models enable solving the biggest challenges of offline RL.

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
