# OpenReview forum: "Advantage-Conditioned Diffusion: Offline RL via Generalization"
_ICLR.cc/2024/Conference — Submitted to ICLR 2024_

### Official Review · Reviewer_CsHK · 2023-10-15

**Soundness:** 2 fair
**Presentation:** 3 good
**Contribution:** 2 fair
**Rating:** 3
**Confidence:** 4

**Summary:**

This paper proposes replacing the traditional reward-to-goal in RvS policy learning with the optimal advantage value as the conditional information. This approach allows for the attainment of the optimal policy by simply providing 0 as the goal information during evaluation, eliminating the need for complex reward-to-goal design since the optimal advantage value is 0. Specifically, this paper employs expectile regression in IQL to train the approximated optimal advantage value, which is then used as the conditional information when training a diffusion policy. A classifier-free guided sampling method is also implemented to enhance the guidance strength for improved performance. In addition, this paper incorporates the test-time value filtering trick from IDQL[1], which involve sampling multiple action candidates and selecting the best one to further enhance optimality. The primary evaluation is conducted on Antmaze tasks. However, the performance gains compared to the base method, IDQL, are marginal.

[1]IDQL: Implicit q-learning as an actor-critic method with diffusion policies, 2023.

**Strengths:**

1. The proposed method smartly circumvents the notoriously challenging reward-to-goal conditional information design in RvS type methods.
2. The proposed method is simple and easy to implement.
3. The paper is well organized.

**Weaknesses:**

1. Limited performance gains. The proposed method only obtains on-par performances compared to the base method, IDQL. The authors argue that IDQL does not incorporate any guidance during training and requires filtering from multiple sampled action candidates to obtain good results. However, the proposed method, ACD-F, still employs the same filtering trick to obtain good results and does not obtains significant performance gains. Furthermore, without this trick, ACD performs severely worse than IDQL.

2. I noticed in a concurrent work [2] that simply training a BC model using diffusion policy can obtain on-par performance with ACD that utilizes additional conditional information. It seems that even without the additional advantage condition, the simplest Diffusion-BC can already perform well. Therefore, it is hard for me to tell whether the introduced conditional information works in this paper.

3. Limited evaluation. This paper only evaluates on Antmaze tasks and only compares against limited baselines, which is hard for me to access the effectiveness the proposed method.

4. Limited ablation studies. Most design choices in this paper are not well elaborated.

5. The desired optimal advantage value is hard to obtain using expectile regression since severe instability will incurr when the expectile value $\rightarrow 1$. This will result in some gap when implementing the method since only an near-optimal advantage value is allowed.

[2] Consistency Models as a Rich and Efficient Policy Class for Reinforcement Learning, 2023.

**Questions:**

N/A

---

> ### Author Response · Authors · 2023-11-17
> **Response to questions**
>
> We thank Reviewer CsHK for their effort in reading our paper and providing detailed feedback that we will use to turn it into a better shape.
>
> Q1. The ablation studies in this paper are limited.
>
> A1. We tried to provide the most important design insights in the paper, which included the choice of architecture (MLP vs DDPM) and different encoding schemes for advantage value. Can the reviewer point out which crucial design choices were not ablated?
>
> Q2. The desired optimal advantage function is difficult to obtain.
>
> A2. We agree but this is not the essence of our work. We resort to IQL as to the method that does the best job at approximating the optimal advantage function from offline data, which has indeed been a long-lasting problem in offline RL (see Kumar et al., 2020, for example). In fact, our method is agnostic to a specific Q-learning algorithm that derives these advantages. Instead, it shows how one can use them to replace traditional policy-optimization steps with generative modeling training.

---

> > ### Comment · Reviewer_CsHK · 2023-11-20
> > **Thanks for the response**
> >
> > Thanks for the reply. However, the authors did not address my concerns. Therefore, I would keep my score unchanged.
> >
> > - In particular, the authors did not address my concerns stated in Weaknesses 1 & 2 that the performance gain is limited.
> > - In addition, the authors did not provide more experimental results on other tasks to show the effectiveness of the proposed method.
> > - Also, the ablation studies that the author mentioned are only conducted on one or two tasks. Therefore, I cannot evaluate whether such design choices also work on other tasks.

---

### Official Review · Reviewer_PrNq · 2023-10-20

**Soundness:** 2 fair
**Presentation:** 2 fair
**Contribution:** 2 fair
**Rating:** 3
**Confidence:** 3

**Summary:**

This work considers using conditional generative models for offline RL, which are conditioned on the advantage function. Then it utilizes IQL for learning approximately optimal advantages for in-distribution actions and Diffusion models for improving the expressive ability. Moreover, this work investigates many design choices, like network architectures, training, training, and sampling steps to test the performance. Finally, this work evaluates ACD and ACD-F in several tasks of D4RL, especially some antmaze tasks.

**Strengths:**

- Utilizing the advantage-conditioned diffusion model as policies for offline RL is interesting.

**Weaknesses:**

- The Experiments Section is too short, due to the ICLR requirement of 9 pages and this article currently only has 8 pages, I hope to see more experiment results and corresponding analyses (See Questions).

- Many techniques of the proposed ACD are based on IDQL, but the performance of ACD is lower than IDQL.

**Questions:**

- In D4RL, there are several locomotion tasks, like HalfCheetah/Walker/Hopper-Medium-Expert/Medium/Medium-Replay, but experiments in Table 4 only test in HalfCheetah-Medium, what is the performance in other locomotion tasks?

- What about the results of ACG in all evaluate tasks?

- The authors have said that " poor results for ACG with scalar advantages can be improved by changing the optimizer from Adam to AdamW ". Is there any ablation study about the performance of Adam and AdamW?

- As the title of this work includes ''Offline RL via Generalization", this work seems to mainly tackle the single task, what is the meaning of "generalization" in this work?

---

> ### Author Response · Authors · 2023-11-17
> **Response to questions**
>
> We thank Reviewer sXjs for their effort in reading our paper and providing detailed feedback that we will use to turn it into a better shape.
>
> Q1. What are the results in other D4RL locomotion tasks? The only given one is HalfCheetah-Medium.
>
> A1. We did not include these results because there was very little difference between our method and IDQL there. For example, our Walker2d-medium was 76.8 (vs IDQL 73.5), and HalfCheetah-medium-replay 44.8 (vs IDQL 46.8).
>
> Q2. Have you conducted ablation studies on the choice of the optimizer (Adam vs AdamW)?
>
> A2. We have not. We only saw the performance improvement in the experiment that we mentioned. We did not run additional ablation to avoid introducing additional components of complexity to our analysis, especially one that is quite orthogonal to our main contribution. Since the majority of algorithms use Adam, we kept it for fair comparison.
>
> Q3. What does “generalization” in the title stand for?
>
> A3. It stands for the ability of neural networks to generalize beyond seen examples. That is, our policy relies on the diffusion model’s understanding of actions with zero advantage, even though individual states may not be paired with such actions in the dataset.

---

> > ### Comment · Reviewer_PrNq · 2023-11-18
> > **Thanks for the reply**
> >
> > I have read the authors' rebuttal and other reviewers' comments. I have kept my scores since I think this article can be improved in many aspects, especially in terms of the organization of the experiment.
> >
> > - In the results of D4RL, it is necessary to include more results about locomotion like half-cheetah/hopper/walker - medium-replay/medium/expert with more analyses about why the proposed method works or performs similarly compared with IDQL.
> >
> > - I appreciate the authors' efforts in analyzing different design choices, which can really help the community. I think these ablation studies (like Table 2 and Table 3) should be evaluated in more settings and discussed more, i.e., which design choices are more important and do they contain any insights, that can make this work more solid.

---

### Official Review · Reviewer_sXjs · 2023-10-30

**Soundness:** 2 fair
**Presentation:** 3 good
**Contribution:** 3 good
**Rating:** 5
**Confidence:** 4

**Summary:**

The authors propose an algorithm for offline RL which uses IQL to train an optimal Advantage function which does not require out of distribution querying for TD updates, and then train a diffusion policy which is conditioned on the advantage. This method of distillation of the policy leverages the expressive capabilities of diffusion models by not requiring some more difficult to optimize weighted form of maximum likelihood such as Advantage Weighted Regression.

**Strengths:**

1) The proposed method is simple and elegant, and I am convinced that it could be an effective method to distill policies from IQL. Advantage Weighted Regression has always been a very flawed method to distill optimal policies from the trained value functions since to obtain an optimal policy, the temperature must be close to 0 which results in a very difficult optimization problem. This approach of advantage conditioned diffusion is much simpler to optimize, while also benefiting from learning a multimodal policy.

2) The paper communicates the motivation and algorithm clearly, and the presentation is good.

3) This work is of high relevance to the offline RL community, which has increasingly become interested in leveraging generative models to frame reinforcement learning as a conditional generative modeling problem.

**Weaknesses:**

1) The experiments are quite weak. While I am fine with offline RL works being evaluated in D4RL as it is the standard, this paper only evaluates on a small subset of the D4RL benchmark. I would like to see evaluations of the method on the other locomotion tasks, Franka Kitchen, and Adroit.

2) AntMaze results are good, but are beaten in the large mazes by the other 2 diffusion based offline RL algorithms DQL and IDQL. While I am fine with the algorithm not necessarily having to perform better than these algorithms, the fact that evaluation has only been done on AntMaze further makes the results look weak.

**Questions:**

1) Could the authors run more extensive experiments across D4RL and some ablations on hyperparameters?

2) Could the authors relate the work to other approaches to diffusion for offline RL that directly try to do return conditioned generative modeling such as Decision Diffuser?

The main proposal is an algorithm which we can only judge based on empirical results, but the experiments are very weak. As such, I am leaning towards rejection of the paper. I am open to increasing the score if presented with a stronger suite of experimental results.

---

> ### Author Response · Authors · 2023-11-17
> **Response to questions**
>
> We thank Reviewer sXjs for their effort in reading our paper and providing detailed feedback that we will use to turn it into a better shape.
>
> Q1. Why did not authors include results from other tasks in D4RL such as Locomotion?
>
> A1. We ran experiments on Locomotion tasks but we did not find these results useful. The majority of methods achieve the same results (see Table 1) and our algorithm is no better.
>
> Q2. How is this algorithm from Decision Diffuser?
>
>
> A2. Decision Diffuser can be viewed as one of the RvS methods, since it conditions the policy on the trajectory return, and as such is captured by prior work that we described. Our method leverages the temporal compositionality of the MDP by employing the advantage function which (approximately) indicates an action’s optimality if it equals zero. On the engineering side of things, Decision Diffuser models the future states. It concatenates K future states and diffuses over the resulting matrix like if it was an image. Instead, we model (diffuse over) the action variable directly.

---

> ### Comment · Reviewer_sXjs · 2023-11-20
> **Thanks for the response**
>
> I appreciate the response by the authors. However, the experimental results are weak in the current draft of the paper and so I will maintain my score. Even if performance is not greatly improved, I feel it is necessary to show evaluations across more tasks in the D4RL benchmark. This is especially important here because the proposed algorithm performs worse than other diffusion based offline RL algorithms even in the few tasks that have been presented.

---

### Official Review · Reviewer_9qGT · 2023-11-04

**Soundness:** 3 good
**Presentation:** 3 good
**Contribution:** 3 good
**Rating:** 6
**Confidence:** 4

**Summary:**

The authors propose an offline reinforcement learning method based on Implicit Q-learning (IQL) and diffusion based generative models. The proposed approach encodes the learned policy as a generative model conditioned on the desired advantage value. This generative model is trained on a dataset of state, action and estimated advantage value triplets with the goal of generating actions conditioned on state and advantage value. The advantage values are estimated using IQL with the stated goal of avoiding maximization of state-action values done in other Q-learning like methods. This is done by replacing the classical maximization, e.g., $\text{max}_a \hat{q}(s,a)$, with an expectile favoring larger values. This expectile is estimated over actions in the dataset in order to avoid out-of-distribution issues.

First, the value of using IQL and advantage conditioned is explored using a simple MLP gaussian policy in three "antmaze" D4RL tasks. This variant is shown to perform better in 2/3 of tasks compared to an actor-critic, a behavior cloning (BC), and a weighted max-likelihood (AWR) method. Different encoding schemes are considered and two "antmaze" tasks. Between a raw encoding (just real-value), a one-hot discretization scheme, the discretization approach performed better but this depends on the optimizer used (Adam vs. AdamW).

Following this, the "full" proposed diffusion model based method, called advantage-conditioned diffusion (ACD), is evaluated. Similarly, different encoding of the advantage values are considered but the authors found simply using the raw value to perform best. The show better performance in the antmaze-medium-play task which corroborates their conclusion.

Finally, a "value filtering" variant (ACD-F) of their method is proposed which instead of generating a single action will sample several actions and pick the highest valued action. Both ACD methods are compared to various baselines on 7 D4RL tasks. Their results show that ACD performs worse than IQL and IDQL, comparable to DQL and better than BC and AWR. Their results also show that ACD-F is comparable to IDQL in all but 2 tasks, and otherwise outperforms all other methods.

**Strengths:**

The main contribution is a nice addition to this idea of value-conditioned policies. The prospect of being able to leverage past and future advances in generative modelling is appealing.

The paper is well written and well structured. The authors do a good job of giving necessary background on relevant offline RL literature.

I quite appreciated the discussion about the different design choices tried, most notably the discussion related to encoding the advantage values when conditioning.

**Weaknesses:**

The empirical results are a little on the weak side. I think the results are sufficient but I would have appreciated some further discussion comparing ACD-F to IDQL. In short, why might I ever consider ACD-F over IDQL? What potential advantages ACD-F might have?

There is little discussion on the experimental setup making reproducibility uncertain. Were baseline results taken from prior work, if so which ones? How were hyperparameters decided? What hyperparameters were chosen (learning rate, $\tau$, $\omega$, the optimizer, etc.)? How many seeds when learning the policy? How many seeds when evaluating the policies?

**Questions:**

p. 4, para before sec 3.3, Do we assume we are operating in an episodic setting (i.e., trajectories always end in terminal states)? If that's the case, this should be stated. Otherwise, how do you properly estimate returns?

p. 5, thm 1, is the optimal policy unique?

p. 5, Fig 1, what is $\tau$?

p. 6, "improving upon the behavior policy in all three tasks", what value does the behavior policy get? Would it make sense to add it to table 1?

p. 6, sec 4.2, how necessary is this additional "unconditional" aspect? Have you tried without it? Have you tried using a separate network for the unconditional part?

p. 7, "while we found these two approaches perform similarly", by what measure do they perform similarly if not by the policy performance?

p. 7, sec 5, what is the runtime of ACD and ACD-F during training and testing? How does it compare to IQL, DQL, and IDQL?

# Minor comments

p. 2, "opem", typo

p. 2, "bahavior", typo

p. 3, "expected total reward", that's the expected *discounted* total reward

p. 4, "substricts", typo

p. 5, "converge to the optimal action", is this really true if we constrain to learn the advantage values to be "in-distribution"?

p. 7, "MPL", typo

p. 7, "sinsusoidal", typo

---

> ### Author Response · Authors · 2023-11-17
> **Response to questions**
>
> We thank Reviewer 9qGT for their effort in reading our paper and providing detailed feedback that we will use to turn it into a better shape.
>
> Q1. Does your method assume the episodic or discounted setting of RL?
>
> A1. Due to the empirical style of the paper, we worked under the usual assumption of practitioners - we work in a discounted setting in which an episode ends in finite time with probability 1. This allows us to employ TD-learning methods, like IQL, to learn action-value functions. The cited RvS works assume the finiteness of episodes too and ignore the discount factor, since their methods relying on supervised learning are agnostic to it.
>
> Q2. Is the optimal policy from Theorem 1 unique?
>
> A2. Thank you for pointing this out. This optimal policy is not unique, since if a few actions have zero advantage function then any probability distribution over them is optimal. We clarified that in the revised version.
>
> Q3. What values does the behavior policy, mentioned on page 6, get in tasks from table 1?
>
> A3. We considered the BC policy as a proxy to behavior policy. We can make it more explicit if the reviewer thinks that would be better.
>
> Q4. How important is the unconditional model from section 4.2?
>
> A4. We believe that we explained why we use this model and classifier-free guidance by saying “In our experiments, we found that this technique can significantly boost performance (e.g., from 43.2 to 56.1 on antmazemedium-play), and throughout the paper we work with ω = 3.”
> Does the reviewer mean something else?
>
> Q5. Page 7: What is meant by “While we found these two approaches perform similarly, they significantly worsened the performance of our policy”?
>
> A5. Thank you for pointing out this writing error. We meant “... worsened the performance of our policy with respect to the raw input”, as seen in Table 3. We corrected it now.

---

### Meta-Review · Area_Chair_6KK2 · 2023-12-12

**Metareview:**

The authors propose an algorithm for offline RL that is based on Implicit Q-learning (IQL) and diffusion-based generative models. Instead of explicitly maximizing the neural function approximator of the critic, they implicitly maximize the value function by the expressive conditional generative models combined with implicit Q-learning backups. They evaluate their proposed method in several tasks of D4RL, especially in some antmaze tasks.

The reviews are split. The positive reviews focus on the good presentation and the nice idea of leveraging past and future advances in generative modeling. The negative reviews focus on the limited evaluation aspect. Specifically, the performance of the proposed method should be evaluated on more tasks such as half-cheetah/hopper/walker rather than only HalfCheetah-Medium task. Moreover, the ablation studies in this paper are also limited, hence most design choices in this paper are not well elaborated. Also, the information gains of the proposed method are not significant IDQL, which casts doubts on the effectiveness of the proposed method. The author did not solve these concerns, hence I tend to recommend rejection.

**Justification For Why Not Higher Score:**

Lacking more experiments like half-cheetah/hopper/walker and more ablation studies. Not significant performance gains compared with IDQL.

**Justification For Why Not Lower Score:**

N/A

---

### Decision · Program_Chairs · 2024-01-16

Reject